# Characterization of the Aroma Profile of Commercial Prosecco Sparkling Wines

Davide Slaghenaufi [1],*, Giovanni Luzzini [1], Matteo Borgato [1], Anita Boscaini [2], Andrea Dal Cin [2], Vittorio Zandonà [2] and Maurizio Ugliano [1]

1   Department of Biotechnology, University of Verona, Via della Pieve 70, 37029 San Pietro Cariano, Italy
2   Masi Agricola, Via Monteleone 26, 37015 Sant'Ambrogio di Valpolicella, Italy
*   Correspondence: davide.slaghenaufi@univr.it

**Abstract:** In this work, the aromatic characterization of commercially available Prosecco wines with a price range between EUR 7 and 13 was carried out. These wines came from three different areas of origin: Valdobbiadene, Asolo and Treviso. Seventy volatile compounds were identified and quantified in the wines. Quantitatively, the wines were mainly characterized by compounds of fermentation origin (alcohols, acids, esters), and $C_6$-alcohols, and to a lesser extent, terpenes, low molecular weight volatile sulfur compounds (VSC), and benzenoids. To determine their impact on the aroma of Prosecco wine, the respective OAVs were calculated. The molecules with higher OAV were ethyl hexanoate, isoamyl acetate, and β-damascenone. More generally, esters, responsible for fruity notes, seemed to play a major role in the aroma of Prosecco wine. Investigation into the possible effect of different production zones indicated 16 significantly different compounds accounting for differences between the various areas of origin of the wines, being mostly VSC, esters and $C_6$-alcohols. A sensory evaluation through a sorting task highlighted the formation of clusters; wine samples were divided into two main groups partially attributable to the areas of origin. From a chemical point of view, cluster A was richer in esters, while cluster B had, on average, higher concentrations of compounds associated with wine aging such as cyclic terpenes, norisoprenoids (TDN and vitispirane), and VSC.

**Keywords:** Prosecco; sparkling wine; volatile compounds; SPME; GC-MS





## 1. Introduction

Prosecco is a white wine produced in northern Italy in the Veneto and Friuli Venezia Giulia regions. It is produced both as a still white wine and as a sparkling wine. The latter type enjoys enormous commercial success all over the world, ranking first in the world among sparkling wines in terms of export volume with 273 million liters, followed by Champagne with 94 million liters [1].

According to the "Denominazione di Origine Controllata" (Controlled Designation of Origin, DOC) regulation, Prosecco sparkling wine is produced using the Glera grape variety, with small amounts of other authorized grape varieties also admitted (<15%). The typical training systems are the Sylvoz and "Doppio capovolto". The maximum production yields are 180 q/ha for the DOC Prosecco, and 135 q/ha for the "Denominazione di Origine Controllata e Garantita" (Controlled and Guaranteed Designation of Origin, DOCG) of Asolo and Valdobbiadene. The secondary fermentation is carried out following the Charmat method, in stainless steel pressurized tanks (autoclave). The second fermentation lasts over a month until an overpressure of about 6 bar is reached. In order to guarantee the desired residual sugar, the refermentation is stopped by refrigerating the wine below zero. After stabilization, the Prosecco wine is bottled in isobaric conditions. Compared with Champenoise methods, consumers have lower expectations towards Charmat method wines [2]. At the same time, the Charmat method is much less complex and costly than

Champenoise, and in some cases, consumer preference in blind tasting has been shown to lean towards Charmat wines. Furthermore, Charmat wines are usually produced more quickly compared with Champenoise wines, which are usually destined for long bottle-aging periods. There is, therefore, great interest in acquiring information in order to assist winemakers in developing production processes adapted to wine characteristics.

An extensive zonation study of Prosecco appellation area has been undertaken in an effort to assess terroirs and cru vineyards [3], resulting in the definition of the high-quality DOCG sub-areas "Asolo", "Conegliano-Valdobbiadene" and "Cartizze".

A volatile chemical profile plays an important role in the quality of sparkling wines, as it determines the olfactory characteristics of the product and, therefore, consumer preferences [4].

Many factors influence the volatile chemical profile of sparkling wines including production method, grape variety, base wine production method, grape origin, yeast strain, aging periods and lees contact [5,6].

Prosecco wines are described with aroma notes of peach, pear, wisteria, flowers, ripe fruit, citrus fruit, spice and sage, with differences due to the sub-areas of grape production [3]. Regarding the chemical composition of Prosecco wine, few works have studied the profile of the volatile compounds responsible for its olfactory bouquet. GC-MS analysis of aroma compounds in Glera grapes showed the presence of glycoconjugated precursors of terpenoids, in particular, geraniol and cis-8-OH-linalool, and also of benzenoids such as phenylethanol and benzyl alcohol, all in the order of a few hundred µg/L [3,7].

Many studies have been undertaken on aspects of consumer preferences, marketing, and economics related to Prosecco, but few studies have analyzed the chemical composition of the wine. Some articles have investigated the mineral composition of wine for authentication purposes [8,9]. This study has two aims: to provide an extensive chemical and sensory characterization of the volatile composition of commercial Prosecco wines; and to investigate the influence of three geographic origins, considered in the PDO regulation, on aroma compounds and sensory characteristics.

## 2. Materials and Methods

### 2.1. Reagents

Octan-2-ol (97%), β-citronellol (95%), 1-hexanol (99%), nerol (≥97%), cis-3-hexenol (98%), trans-3-hexenol (97%), ethyl acetate (99%), vanillin (99%), linalool (97%), α-terpineol (90%), terpinen-4-ol (≥95%), geraniol (98%), terpinolene (≥85%), linalool oxide (≥97%), p-cymene (99%), γ-terpinene (≥97%), 1,4-cineole(≥98.5%), 1,8-cineole (99%), β-damascenone (≥98%), isoamyl alcohol (98%), ethyl butanoate (99%), limonene (97%), benzyl alcohol (≥99%), 2-phenylethanol (≥99%), ethyl 3-methyl butanoate (≥98%), isoamyl acetate (≥95%), ethyl lactate (≥98%), ethyl hexanoate (≥95%), n-hexyl acetate (≥98%), ethyl octanoate (≥98%), ethyl decanoate (≥98%), hexanoic acid (≥99%), octanoic acid (≥98%), α-phellandrene (95%), phenylethyl acetate (99%), p-menthane-1,8-diol (97%), ethyl vanillate (99%), 3-methylbutanoic acid (99%), 1-butanol (≥99%), methyl-vanillate (99%), vinyl guaiacol (≥98%), benzaldehyde (≥99%), nerolidol (98%), bisabolol (≥93%), methyl salicylate (≥99%), rose oxide (≥98%), β-pinene(99%), 3-carene (≥90%), α-terpinen (≥95%), ethyl cinnamate (99%), cis-2-hexenol (95%), ethyl 2-hydroxybutyrate (99%), β-Myrcene (≥90%), ethyl 3-hydroxybutyrate (≥98%), carbon disulfide (≥99%), dimethyl sulfide (≥99%), methionol (98%), diethyl sulfide (98%), dimethyl disulfide (≥98%), ethyl thioacetate (≥98%), diethyldisulfide (99%), and dimethyl trisulfide (≥98%) were supplied by Sigma Aldrich (Milan, Italy). 1,1,6-Trimethyl-1.2-dihydronaphtalene (TDN) with 80% purity was supplied by Synchem UG & Co. (Felsberg, Germany). Sodium chloride (≥99.5%) was provided by Sigma Aldrich (Milan, Italy). Methanol (≥99.8%), and dichloromethane (≥99.8%) were supplied by Honeywell (Seelze, Germany).

## 2.2. Wine Samples

For this study, twenty-four commercial Prosecco sparkling wines were purchased in a price range between EUR 7 and 13, corresponding to the typical average price of quality Prosecco wine available in the local market. The wines were not millesimé, therefore, could have been produced from harvests of different vintages. The samples were classified as brut and extra dry and came from three different denominations—DOC Treviso, DOCG Asolo and DOCG Conegliano Valdobbiadene—as shown in Table 1.

**Table 1.** Prosecco wine sample, appellation and sweetness classification.

| Sample | PDO | Sugar Level [1] |
|:---:|:---:|:---:|
| 1 | Treviso | Extra dry |
| 2 | Valdobbiadene | Extra dry |
| 3 | Valdobbiadene | Brut |
| 4 | Valdobbiadene | Extra dry |
| 5 | Valdobbiadene | Brut |
| 6 | Valdobbiadene | Extra dry |
| 7 | Valdobbiadene | Extra dry |
| 8 | Valdobbiadene | Extra dry |
| 9 | Valdobbiadene | Brut |
| 10 | Valdobbiadene | Extra dry |
| 11 | Valdobbiadene | Extra dry |
| 11 | Valdobbiadene | Extra dry |
| 12 | Treviso | Brut |
| 13 | Treviso | Brut |
| 14 | Valdobbiadene | Extra dry |
| 15 | Valdobbiadene | Extra dry |
| 16 | Valdobbiadene | Brut |
| 17 | Treviso | Extra dry |
| 18 | Valdobbiadene | Extra dry |
| 19 | Valdobbiadene | Extra dry |
| 20 | Treviso | Extra dry |
| 21 | Asolo | Extra dry |
| 22 | Asolo | Extra dry |
| 23 | Asolo | Extra dry |
| 24 | Asolo | Extra dry |

[1] Brut and extra dry refer to residual sugar concentrations of 0–12 g/L and 12–17 g/L, respectively.

## 2.3. Analysis of Volatile Sulfur Compounds

Low molecular weight sulfur compounds, were analyzed by SPME-GC-MS as described by Slaghenaufi et al. [10]. In order to prevent compounds volatilization, wine samples were kept at 4 °C for 24 h prior to analysis. Samples were prepared by adding 100 μL of DMS-d6 internal standard (2 mg/L in ethanol) to 10 mL of wine placed in a 20 mL glass vial together with 3 g NaCl. Samples were then kept at 4 °C until SPME extraction. SPME extraction and injection were performed using an automatic sampler Gerstel MPS3 (Müllheim/Ruhr, Germany). Prior to extraction, samples were equilibrated for 1 min at 40 °C, then a polydimethylsiloxane-divinylbenzene fiber (PDMS/DVB) (Supelco, Bellafonte, PA, U.S.A.) was exposed to the sample headspace for 30 min. VSCs were desorbed in the injector port at 270 °C for 2 min in splitless mode. The analysis was performed using a HP 7890A (Agilent Technologies, Santa Clara, CA, USA) gas chromatograph coupled to a 5977B quadrupole mass spectrometer. Chromatographic separation was achieved by using a DB-WAX UI capillary column (30 m × 0.25, 0.25 μm film thickness, Agilent Technologies) and helium (6.0 grade) as carrier gas at 1.2 mL/min of constant flow rate. The column temperature started at 35 °C for 5 min, then increased to 90 °C at 5 °C/min, and then to 260 °C at 15 °C/min, maintained for 2 min. A mass spectrometer was equipped with an electron impact ionization source (EI) (70 eV). The transfer line, source and quadrupole temperatures were set at 200, 250 and 150 °C, respectively. Mass spectra were acquired

in SIM mode. In order to quantify each compound, calibration lines were prepared by adding each analyte in white wine at 7 different concentration levels. Each level was analyzed as a sample following the above procedure. Using Chemstation software (Agilent Technologies, Inc.), the calibration curves were determined by linear regression of the ratio between the concentration of the added analyte and the concentration of the internal standard in the wine, versus the ratio of the peak area of the analyte and the peak area of the internal standard.

### 2.4. Analysis of Terpenoids and Norisoprenoids

Terpenes and norisoprenoids were analyzed using SPME extraction coupled with GC-MS analysis as described by Slaghenaufi et al. [11]. An aliquot of deionized water (5 mL) and of wine sample (5 mL) were added to a 20 mL vial, 3 g of NaCl and 5 μL of internal standard 2-octanol (4.2 mg/L in ethanol). Sampling and injection were performed using a Gerstel MPS3 auto sampler (Müllheim/Ruhr, Germany). Samples were kept for 1 min at 40 °C, and then SPME extraction was performed by placing a 50/30 μm DVB/CAR/PDMS (divinylbenzene–carboxen–polydimethylsiloxane) fiber (Supelco, Bellafonte, PA, U.S.A.) in the vial sample headspace for 60 min. Injection was undertaken in splitless mode by desorbing the SPME fiber into the injection port of an HP 7890A (Agilent Technologies) gas chromatograph coupled to a 5977B mass spectrometer. Chromatographic separation was performed using a polar capillary column DB-WAX UI (30 m × 0.25, with a film thickness of 0.25 μm, Agilent Technologies), inside of which a constant helium flow of 1.2 mL/min was maintained. The purity of helium was grade 6.0. The separation took place with the following temperature gradient: the starting temperature of 40 °C was maintained for 3 min, increased at a rate of 4 °C/min until reaching 230 °C, and this temperature was then maintained for 20 min. The ion source was set at a temperature of 250 °C and operated under electron ionization (EI) with a potential of 70 eV. The quadrupoles were kept at a temperature of 150 °C. The acquisition mode was synchronous SCAN ($m/z$ 40–200) and single ion monitoring (SIM). The order of analysis of the samples was random.

### 2.5. Analysis of Major Volatile Compounds

For quantification of alcohols, esters, fatty acids, and benzenoids were extracted using SPE and then analyzed by GC-MS as described by Slaghenaufi et al. (2020) [12]. Before extraction, 50 mL of wine sample was diluted with deionized water (50 mL), and added with 100 μL of internal standard 2-octanol (4.2 mg/L in ethanol). SPE cartridge BOND ELUT-ENV, (Agilent Technologies, Santa Clara, CA, U.S.A.) was activated by eluting 20 mL of the following solvents, in the following order: dichloromethane, methanol and water. Then the entire sample was loaded by percolating it through the SPE cartridge and then washing with 15 mL of water. A volume of 10 mL of dichloromethane was used to elute volatile compounds. The organic phase was then concentrated to 200 μL using a nitrogen stream. The sample was then ready for GC injection. The gas chromatograph used was an HP 7890A (Agilent Technologies), while the mass spectrometer was a single quadrupole 5977B analyzer. The capillary column used was a DB-WAX UI (30 m × 0.25, with a film thickness of 0.25 μm, Agilent Technologies), and using helium as carrier gas at a constant flow of 1.2 mL/min. The injection of 2 L of sample extract into the GC-MS system was conducted using an auto sampler Gerstel MPS3 (Müllheim/Ruhr, Germany). The inlet was configured in splitless mode at a temperature of 250 °C. The GC oven temperature schedule was set at 40 °C for 3 min, then increased at 4 °C/min to 230 °C. The final temperature was maintained for 20 min. The mass spectrometer operated with the following parameters: the ionization was in EI mode with a potential of 70 eV; the source and quadrupole temperature parameters were 250 °C and 150 °C, respectively; and the analysis was conducted in SIM mode. Samples were analyzed in random order.

For quantification, a calibration curve was obtained for each analyzed compound. Seven different analyte concentrations were prepared in triplicate in a wine-like solution

(12% $v/v$ ethanol, 3.5 g/L tartaric acid, pH 3.5). Each point was analyzed using the same SPE extraction and GC-MS analysis procedure described above for the wine samples.

### 2.6. Standard Enological Analyses

Acetic acid, total acidity (expressed as g of tartaric acid), acetaldehyde, polyphenols, free $SO_2$, total $SO_2$, tartaric acid, malic acid, lactic acid, sugar, glycerol, and yeast assimilable nitrogen (YAN = sum of ammonia and primary ammino acid-derived nitrogen) were analyzed using a Biosystems Y15 multiparametric analyzer (Sinatech, Fermo, Italy). pH was evaluated with a Crison Basic 20+ pH meter (Barcelona, Spain). Ethanol was analyzed using an FTIR wine analyzer Lyza 5000 (Anton Paar, Graz, Austria).

### 2.7. Sorting Task Analysis

The procedure described by Alegre et al. (2017) [13] was used to perform the sorting task analysis. A panel was formed by wine experts according to the definition given by Parr et al. (2002) [14] consisting of 6 male and 6 female member researchers or academic staff regularly participating in wine sensory evaluation. Wine samples were stored at 16 °C. Samples were removed from the cold room one hour before the sorting task. Wines were then poured (20 mL) into ISO glasses, covered with Petri dishes, labelled with a random 3-digit code and served in random order to each panelist. The judges had to evaluate the wines only via orthonas, and to group the wines that presented olfactory similarities. There were no limits on the number of groups, and they were not asked to provide descriptors or other indications.

### 2.8. Statistical Analyses

Hierarchical cluster analysis (HCA) of sensory data, Mann–Whitney and Kruskal–Wallis tests were performed using XLSTAT 2022 (Addinsoft SARL, Paris, France).

## 3. Results and Discussion

### 3.1. General Composition of Prosecco Wines

The data of the basic oenological parameters are reported in Table 2. Although the wines were from different producers and purchased in the same local supermarket, a certain homogeneity of the data was observed. Tartaric acid, acetic acid, lactic acid, residual sugars, free $SO_2$, $NH_4$ and amino nitrogen showed relative standard deviation variations greater than 20%.

**Table 2.** Base enological parameters.

|  | Min | Max | Mean | Standard Deviation | RSD [3] (%) |
|---|---|---|---|---|---|
| Ethanol ($v/v$) | 11.06 | 12.13 | 11.38 | 0.27 | 2.41 |
| Free $SO_2$ (mg/L) | 8.0 | 41.8 | 20.4 | 8.1 | 40 |
| Total $SO_2$ (mg/L) | 80 | 158 | 124 | 22 | 18 |
| Tartaric acid (g/L) | 0.8 | 2.0 | 1.4 | 0.30 | 21 |
| Acetic acid (g/L) | 0.0 | 0.4 | 0.2 | 0.08 | 51 |
| Malic acid (g/L) | 1.3 | 2.8 | 2.2 | 0.37 | 17 |
| Lactic acid (g/L) | 0.0 | 0.6 | 0.2 | 0.21 | 122 |
| Total acidity (g/L) | 5.2 | 8.9 | 6.0 | 0.94 | 16 |
| Sugar [4] (g/L) | 8.4 | 19 | 13 | 2.9 | 22 |
| Acetaldehyde (mg/L) | 26 | 69 | 58 | 8.9 | 15 |
| Glycerol (g/L) | 3.8 | 5.2 | 4.6 | 0.43 | 9.3 |
| $NH_4$ (mg/L) | <14 | 301 | 48 | 71 | 147 |
| PAN [1] (mg/L) | 12 | 112 | 36 | 19 | 53 |
| YAN [2] (mg/L) | 12 | 352 | 85 | 81 | 96 |
| Total polyphenols (mg/L) | 142 | 242 | 185 | 25 | 14 |

[1] PAN: primary amino nitrogen; [2] YAN: yeast assimilable nitrogen (YAN = $NH_4$ + PAN); [3] RSD: relative standard deviation; [4] brut, sugars < 12 g/L; extra dry, 12 g/L < sugars < 17 g/L.

The wines were characterized by low concentrations, compared with their odor threshold, of acetaldehyde, lactic acid and acetic acid, the latter in all cases well below the legal limit, indicating that the fermentations in general were regular and without issue or microbiological spoilage. The low values or even absence of lactic acid indicated that the wines had not undergone malolactic fermentation. The wines showed different values of free $SO_2$, on average 20 mg/L. These values seemed quite low and suggested that the wines could undergo oxidation or the development of microorganisms.

### 3.2. Volatile Compounds in Prosecco Wines

A total of 70 volatile compounds were identified and quantified in the wine samples, including 5 alcohols, 4 $C_6$-alcohols, 3 acetate esters, 11 ethyl esters, 3 acids, 23 terpenes, 2 sesquiterpenes, 6 norisoprenoids, 7 volatile sulfur compounds (VSC), and 6 benzenoids (Table 3).

**Table 3.** Volatile compounds' minimum concentration (min), maximum concentration (max), and mean concentration of all samples, and relative standard deviation (RSD (%)) in wine samples. Mean value for each production is with significance among the three zones.

| Compound | Min | Max | Mean | RSD | Asolo | Treviso | Valdobbiadene | *p*-Value |
|---|---|---|---|---|---|---|---|---|
| | (µg/L) | (µg/L) | (µg/L) | (%) | (µg/L) | (µg/L) | (µg/L) | |
| Carbon disulfide | 7.43 | 23 | 15 ± 3.77 | 24.97 | 19.93 ± 2.83 b | 13.25 ± 3.61 a | 14.39 ± 3.08 a | 0.016 |
| Dimethyl sulfide | 2.85 | 17.58 | 7.07 ± 3.57 | 50.55 | 9.91 ± 2.31 b | 7.93 ± 5.51 a | 6.74 ± 2.72 ab | 0.062 |
| Diethyl sulfide | 0.21 | 2.55 | 1.21 ± 0.74 | 60.71 | 2.45 ± 0.1 b | 1.27 ± 0.58 a | 0.9 ± 0.48 a | 0.004 |
| Dimethyl disulfide | 0.17 | 4.17 | 1.63 ± 1.34 | 82.34 | 3.17 ± 0.48 b | 2.41 ± 1.39 a | 1.17 ± 1 ab | 0.004 |
| Ethyl thioacetate | 2.56 | 29.6 | 16.2 ± 9.03 | 55.55 | 23.87 ± 8.46 b | 20.36 ± 7.39 a | 12.61 ± 8.26 ab | 0.025 |
| Diethyl disulfide | 0.04 | 0.95 | 0.26 ± 0.28 | 107.12 | 0.77 ± 0.28 b | 0.25 ± 0.14 a | 0.15 ± 0.11 a | 0.006 |
| Dimethyl trisulfide | 0.06 | 3.81 | 0.75 ± 1.1 | 147.74 | 0.18 ± 0.14 a | 1.61 ± 1.48 a | 0.79 ± 1 a | 0.116 |
| Sum of VSC | 18.84 | 66.8 | 42.27 ± 11.69 | 27.65 | 60.28 ± 5.56 a | 47.07 ± 2.04 a | 36.76 ± 8.47 a | 0.001 |
| 1-Butanol | 52.2 | 135 | 86.6 ± 22.14 | 25.55 | 87.8 ± 30.5 a | 78.37 ± 17.54 a | 88.37 ± 22.06 a | 0.659 |
| Isoamyl alcohol | 73,444 | 111,968 | 92,986 ± 9737 | 10.47 | 95,458 ± 14,929 a | 87,321 ± 11,741 a | 92,917 ± 7335 a | 0.446 |
| Phenylethyl alcohol | 3281 | 7868 | 5402 ± 1156 | 21.41 | 5763.61 ± 1387.04 b | 4326.85 ± 852 b | 5515 ± 1027 a | 0.056 |
| Benzyl alcohol | 27.4 | 270 | 75 ± 56.06 | 74.74 | 70.03 ± 25.74 a | 97.76 ± 97.45 a | 68.12 ± 45.53 a | 0.839 |
| Methionol | 112.07 | 326 | 191 ± 60.8 | 31.78 | 230.44 ± 74.48 a | 181.36 ± 84.07 a | 181.44 ± 48.42 a | 0.334 |
| Sum of higher alcohol | 76,923 | 118,681 | 98,655 ± 10,677 | 10.82 | 101,523 ± 16,177 a | 91,927 ± 12,418 a | 98,682 ± 8143 a | 0.409 |
| 1-Hexanol | 644 | 1365 | 930 ± 172 | 18.6 | 1057.83 ± 232.49 a | 913.21 ± 88.58 a | 906.77 ± 172.35 a | 0.405 |
| *cis*-3-Hexen-1-ol | 9.84 | 22.22 | 15.47 ± 3.3 | 21.31 | 16.64 ± 2.87 a | 15.54 ± 2.53 a | 14.98 ± 3.71 a | 0.67 |
| *trans*-3-Hexen-1-ol | 103 | 198 | 141 ± 24.69 | 17.48 | 139.95 ± 28.46 a | 136.64 ± 30.45 a | 140.63 ± 23.48 a | 0.965 |
| *cis*-2-Hexen-1-ol | 8.94 | 23.87 | 13.13 ± 3.12 | 23.77 | 13.66 ± 2.71 a | 12.52 ± 1.23 a | 13.05 ± 3.71 a | 0.774 |
| Sum of C6 alcohols | 792 | 1527 | 1099 ± 179 | 16.31 | 1228.08 ± 221.31 a | 1077.91 ± 61.62 a | 1075.43 ± 188.9 a | 0.369 |
| Isoamyl acetate | 60.4 | 5199 | 1573 ± 1099 | 69.85 | 1547.46 ± 395.13 a | 1242.13 ± 884.88 a | 1595.01 ± 1293.81 a | 0.681 |
| n-Hexyl acetate | 1.44 | 179 | 71.07 ± 44.32 | 62.37 | 70.55 ± 27.01 a | 66.33 ± 41.7 a | 68.76 ± 50.63 a | 0.989 |
| Phenethyl acetate | 25.6 | 579 | 152.68 ± 105 | 69.35 | 158.88 ± 33.93 a | 121.32 ± 59.53 a | 153.22 ± 129.29 a | 0.543 |
| Sum of acetates | 88.1 | 5958 | 1797 ± 1240 | 68.99 | 1776.88 ± 422.59 a | 1429.77 ± 982.87 a | 1816.99 ± 1465.12 a | 0.682 |
| Ethyl acetate | 2.75 | 113 | 52.1 ± 22.94 | 44.03 | 49.09 ± 15.79 a | 46.19 ± 25.39 a | 55.32 ± 24.57 a | 0.82 |
| Ethyl butanoate | 60.85 | 425 | 270 ± 66.05 | 24.39 | 257.95 ± 144.68 a | 262.89 ± 32.2 a | 275.61 ± 47.71 a | 0.673 |
| Ethyl hexanoate | 829 | 1730 | 1117 ± 233.11 | 20.86 | 1293.86 ± 123.66 a | 1065.06 ± 262.5 a | 1075.86 ± 235.59 a | 0.128 |
| Ethyl octanoate | 416.65 | 1142.8 | 740 ± 161.63 | 21.82 | 983.27 ± 111.04 b | 607.3 ± 75.59 a | 709.35 ± 122.66 a | 0.002 |
| Ethyl decanoate | 22.32 | 165.54 | 90.1 ± 33.56 | 37.21 | 111.1 ± 9.82 a | 78.82 ± 28.07 a | 85.22 ± 37.81 a | 0.147 |
| Sum of ethyl esters of straight-chain fatty acids | 1570 | 3282 | 2271 ± 416 | 18.33 | 2695.26 ± 324.96 a | 2060.27 ± 335.26 a | 2201.36 ± 401.13 a | 0.036 |
| Ethyl 2-methylbutyrate | 0.02 | 27.67 | 8.65 ± 7.03 | 81.31 | 7.35 ± 6.32 a | 8.65 ± 7.1 a | 9.76 ± 7.59 a | 0.965 |

**Table 3.** *Cont.*

| Compound | Min | Max | Mean | RSD | Asolo | Treviso | Valdobbiadene | *p*-Value |
|---|---|---|---|---|---|---|---|---|
| | (µg/L) | (µg/L) | (µg/L) | (%) | (µg/L) | (µg/L) | (µg/L) | |
| Ethyl 3-methylbutanoate | 0.1 | 51.18 | 16.82 ± 12.2 | 72.49 | 14.41 ± 12.97 a | 16.78 ± 12.54 a | 18.82 ± 12.7 a | 0.937 |
| Ethyl 3-hydroxybutyrate | 0 | 88.14 | 65.49 ± 19.36 | 29.55 | 40.27 ± 33.42 a | 67.09 ± 11.02 a | 71.62 ± 10.84 a | 0.184 |
| Ethyl 2-hydroxyhexanoate | 0.01 | 1.8 | 0.98 ± 0.37 | 37.81 | 1.17 ± 0.46 a | 1.04 ± 0.26 a | 0.95 ± 0.38 a | 0.689 |
| Sum of ethyl esters of branched acid | 14.14 | 143 | 91.9 ± 25.08 | 27.28 | 63.19 ± 33.36 a | 93.56 ± 23.28 a | 101.15 ± 18.67 a | 0.067 |
| Ethyl cinnamate | 0.01 | 11.79 | 5.62 ± 2.72 | 48.34 | 4.87 ± 2.78 a | 4.63 ± 2.76 a | 6.21 ± 2.74 a | 0.649 |
| Ethyl lactate | 1669 | 10,588 | 4468 ± 2190 | 49.02 | 3547 ± 1476 a | 4490 ± 1594 a | 4778 ± 2526 a | 0.747 |
| Sum of other esters | 1670 | 10,597 | 4474 ± 2191 | 48.98 | 3552 ± 1478 a | 4494 ± 1594 a | 4784 ± 2528 a | 0.747 |
| 3-Methylbutanoic acid | 173 | 352 | 259 ± 41.6 | 16.05 | 283.5 ± 46.8 a | 231.65 ± 37.33 a | 264.3 ± 39.1 a | 0.221 |
| Hexanoic acid | 4384 | 7647 | 6227 ± 755 | 12.14 | 6614 ± 760 b | 5407 ± 765 b | 6311 ± 577 a | 0.035 |
| Octanoic acid | 140 | 10,211 | 8820 ± 1989 | 22.55 | 7166 ± 4701 a | 8499 ± 771 a | 9266 ± 674 a | 0.111 |
| Sum of fatty acids | 7081 | 18,020 | 15,307 ± 2242 | 14.65 | 14,065 ± 4756 a | 14,139 ± 1508 a | 15,842 ± 1132 a | *0.08* |
| *cis*-Linalool oxide | 1.23 | 25.06 | 6.46 ± 5.57 | 86.12 | 3.71 ± 2.22 a | 7 ± 4.46 a | 7.47 ± 6.45 a | 0.52 |
| *trans*-Linalool oxide | 0.68 | 13.14 | 3.33 ± 2.57 | 77.27 | 2.19 ± 1.26 a | 3.68 ± 2.22 a | 3.77 ± 2.95 a | 0.332 |
| Linalool | 0.85 | 88.08 | 11.8 ± 17.1 | 144.5 | 9.22 ± 5.24 a | 6.04 ± 3.64 a | 13.71 ± 21.29 a | 0.571 |
| Terpinen-1-ol | <LOQ | 1.27 | 0.12 ± 0.25 | 201.8 | 0.12 ± 0.09 a | 0.07 ± 0.05 a | 0.14 ± 0.32 a | 0.542 |
| Terpinen-4-ol | 0.04 | 3.16 | 1.22 ± 0.65 | 53.72 | 1.28 ± 0.34 a | 0.89 ± 0.5 a | 1.31 ± 0.75 a | 0.526 |
| Ho-trienol | <LOQ | 0.62 | 0.06 ± 0.13 | 198.39 | 0.03 ± 0.02 a | 0.02 ± 0.02 a | 0.09 ± 0.16 a | 0.674 |
| α-Terpineol | 0.21 | 13.08 | 6.19 ± 2.87 | 46.29 | 6.81 ± 2.33 a | 4.64 ± 2.61 a | 6.46 ± 3.05 a | 0.481 |
| Nerol | 0.03 | 29.74 | 2.34 ± 6.19 | 264.89 | 1.67 ± 1.72 b | 0.31 ± 0.23 ab | 3 ± 7.74 a | 0.096 |
| Geraniol | 0.05 | 7.31 | 2.56 ± 1.84 | 72.07 | 4.14 ± 0.84 b | 2.9 ± 2.99 a | 2.14 ± 1.35 ab | 0.088 |
| β-Citronellol | 0.11 | 7.95 | 2.64 ± 1.89 | 71.44 | 1.92 ± 0.79 a | 1.47 ± 1.04 a | 3.07 ± 2.1 a | 0.17 |
| α-Phellandrene | <LOQ | 0.32 | 0.12 ± 0.08 | 71.32 | 0.25 ± 0.08 b | 0.07 ± 0.04 a | 0.1 ± 0.06 a | 0.012 |
| 1,4-Cineole | 0.03 | 0.46 | 0.17 ± 0.1 | 61.54 | 0.18 ± 0.04 a | 0.17 ± 0.12 a | 0.17 ± 0.11 a | 0.579 |
| 1,8-Cineole | 0.04 | 0.39 | 0.49 ± 1.64 | 333.76 | 0.11 ± 0.03 a | 0.23 ± 0.15 a | 0.16 ± 0.08 a | 0.452 |
| Limonene | 0.05 | 14.78 | 0.83 ± 2.98 | 359.05 | 0.45 ± 0.17 b | 0.22 ± 0.17 a | 1.07 ± 3.78 ab | 0.037 |
| γ-Terpinene | 0.06 | 5.74 | 2.12 ± 1.45 | 68.52 | 3.73 ± 1.68 b | 1.76 ± 1.42 ab | 1.74 ± 1.19 a | 0.095 |
| p-Cymene | 0.02 | 0.4 | 0.15 ± 0.08 | 55.74 | 0.23 ± 0.07 b | 0.15 ± 0.09 a | 0.14 ± 0.08 ab | 0.047 |
| Terpinolene | 0.04 | 0.34 | 0.17 ± 0.08 | 50.7 | 0.24 ± 0.08 a | 0.15 ± 0.07 a | 0.15 ± 0.08 a | 0.186 |
| p-Menthane-1,8-diol | <LOQ | 16.46 | 3.54 ± 3.93 | 110.9 | 2.11 ± 1.86 a | 3.73 ± 2.67 a | 4.12 ± 4.67 a | 0.704 |
| α-Terpinene | <LOQ | 0.14 | 0.06 ± 0.03 | 55.29 | 0.1 ± 0.03 b | 0.04 ± 0.02 a | 0.05 ± 0.03 a | 0.021 |
| β-Myrcene | 0.08 | 3.7 | 1.25 ± 1.06 | 85.14 | 2.64 ± 0.93 b | 0.6 ± 0.33 a | 1.06 ± 0.95 a | 0.025 |
| 3-Carene | 0.02 | 0.23 | 0.1 ± 0.05 | 54.79 | 0.15 ± 0.05 b | 0.13 ± 0.08 ab | 0.07 ± 0.03 a | 0.012 |
| β-Pinene | 0.01 | 2.55 | 0.22 ± 0.5 | 229.79 | 0.27 ± 0.09 b | 0.08 ± 0.03 a | 0.24 ± 0.64 a | 0.037 |
| Rose oxide | <LOQ | 0.04 | 0.02 ± 0.01 | 65.79 | 0.03 ± 0.01 b | 0.01 ± 0.01 ab | 0.02 ± 0.01 a | 0.038 |
| Sum of monoterpenoids | 12.9 | 142.2 | 46 ± 24.7 | 53.7 | 41.54 ± 7.46 a | 34.37 ± 13.12 a | 50.24 ± 27.59 a | 0.499 |
| β-Damascenone | 0.07 | 5.23 | 1.52 ± 1.15 | 75.72 | 1.64 ± 0.81 a | 1.03 ± 0.73 a | 1.59 ± 1.34 a | 0.41 |
| Vitispirane 1 | 0.01 | 4.59 | 1.07 ± 0.97 | 90.68 | 1.55 ± 0.82 a | 1.67 ± 1.76 a | 0.98 ± 0.45 a | 0.143 |
| Vitispirane 2 | 0.01 | 2.32 | 0.74 ± 0.51 | 69.32 | 1.04 ± 0.66 a | 0.91 ± 0.86 a | 0.71 ± 0.25 a | 0.265 |
| 1-(2,3,6-Trimethylphenyl)-buta-1,3-diene (TPB) | <LOQ | 0.15 | 0.04 ± 0.04 | 97.03 | 0.09 ± 0.05 b | 0.01 ± 0.01 a | 0.03 ± 0.03 a | 0.016 |
| 1,1,6-Trimethyl-1,2-dihydronapthalene (TDN) | <LOQ | 7.92 | 2.01 ± 1.84 | 91.96 | 3.27 ± 1.75 b | 3 ± 3.06 a | 1.75 ± 0.99 ab | 0.094 |
| 3-Oxo-α-ionol | 0.65 | 4.32 | 2.58 ± 0.87 | 33.66 | 2.46 ± 1.5 a | 2.71 ± 0.85 a | 2.53 ± 0.73 a | 0.867 |
| Sum of norisoprenoids | 1.76 | 16.74 | 6.43 ± 3.23 | 50.26 | 8.4 ± 1.81 a | 8.3 ± 5.5 a | 6 ± 2.01 a | 0.064 |
| Nerolidol | 0.54 | 4.07 | 1.71 ± 1.08 | 63.43 | 1.33 ± 0.58 a | 1.71 ± 1.4 a | 1.74 ± 1.12 a | 0.769 |
| Bisabolol | 0.58 | 4.98 | 1.44 ± 0.83 | 57.82 | 1.21 ± 0.55 a | 1.21 ± 0.57 a | 1.6 ± 0.96 a | 0.199 |
| Sum of sesquiterpenoids | 1.45 | 5.89 | 3.15 ± 1.26 | 40.02 | 2.55 ± 0.99 a | 2.91 ± 1.4 a | 3.34 ± 1.28 a | 0.319 |

**Table 3.** *Cont.*

| Compound | Min | Max | Mean | RSD | Asolo | Treviso | Valdobbiadene | *p*-Value |
|---|---|---|---|---|---|---|---|---|
| | (µg/L) | (µg/L) | (µg/L) | (%) | (µg/L) | (µg/L) | (µg/L) | |
| 4-Vinylguaiacol | 3.42 | 13.06 | 7.21 ± 2.11 | 29.29 | 4.81 ± 1.54 a | 7.91 ± 1.89 ab | 7.6 ± 1.95 b | 0.044 |
| Vanillin | 2.07 | 25.42 | 3.66 ± 4.71 | 128.6 | 9.01 ± 11.11 a | 2.64 ± 0.32 b | 2.59 ± 0.33 b | 0.774 |
| Methyl-vanillate | 3.07 | 6.05 | 4.17 ± 0.66 | 15.85 | 3.74 ± 0.48 a | 4.28 ± 1 a | 4.21 ± 0.57 a | 0.434 |
| Ethyl-vanillate | 0.49 | 3.08 | 1.77 ± 0.58 | 32.95 | 1.64 ± 0.89 a | 1.64 ± 0.36 a | 1.82 ± 0.58 a | 0.85 |
| Methyl salycilate | 0.31 | 6.69 | 3.39 ± 1.39 | 41.03 | 4.23 ± 1.65 a | 2.49 ± 1.27 a | 3.42 ± 1.28 a | 0.246 |
| Benzaldehyde | <LOQ | 11.82 | 5.04 ± 4.06 | 80.52 | 0.94 ± 0.82 a | 4.37 ± 3.77 a | 6.18 ± 4 a | 0.013 |
| Sum of benzenoids | 13.41 | 45.23 | 25.24 ± 7.81 | 30.95 | 24.36 ± 14.53 a | 23.34 ± 7.07 a | 25.82 ± 6.2 a | 0.537 |

Values in the same row with different letters indicate statistically significant differences, *p* < 0.1.

### 3.2.1. Higher Alcohols

In terms of quantity, higher alcohols represent the class of volatile compounds showing higher concentrations. These compounds are produced during alcoholic fermentation by yeast from either sugars or amino acids (Ehrlich pathway). The most abundant in the studied wines were isoamyl alcohol and 2-phenylethanol. 2-Phenylethanol and methionol are described as rose and cooked potatoes, respectively, while all other higher alcohols are generally characterized by solvent odor. However, isoamyl alcohol was the only higher alcohol among those analyzed which exceeded its olfactory threshold, thus, contributing to the aroma of the analyzed wine samples. The higher alcohol concentrations in the samples did not vary much between samples, e.g., isoamyl alcohol and phenylethyl alcohol showed variability of 10 and 21%, respectively (as relative standard deviation, RSD).

### 3.2.2. Acids

The fatty acids represent the second family of volatile compounds in terms of concentration. They are characterized by cheesy and sweaty notes and are by-products of alcoholic fermentation. They are considered to be contributors to the vinous character of the wine. It was observed that the concentration of fatty acids in the samples was quite stable, with variations in the order of 12–22%. The fatty acid present in the highest concentration was octanoic acid, on average at 8.8 mg/L, followed by hexanoic acid at 6.2 mg/L. All three fatty acids analyzed were present in the wines at concentrations well above their olfaction thresholds and could, therefore, actively contribute to the wine samples' aroma.

### 3.2.3. Esters

Esters are a class of compounds characterized by fruity notes and are produced by yeast during fermentation. Their formation is influenced by various factors linked both to the composition of the grapes (°Brix, nitrogen content) and to technological factors such as fermentation temperature, clarity of must, and yeast strain. On average, the major ester was ethyl lactate, followed by isoamyl acetate, ethyl hexanoate and ethyl octanoate. Variations in esters content could depend on a number of factors including yeast strain for primary and secondary fermentation, nutrient profile of must and base wine, turbidity, and fermentation temperature [15]. The data showed that the ethyl esters varied relatively little, with ethyl butanoate, ethyl hexanoate and ethyl octanoate showing RSD (relative standard deviation) values of 24, 21 and 22%, respectively. Conversely, the acetic esters showed RSD variations greater than 60%. These variations could have been due to the age of the samples or the base wines used for refermentation. In fact, it is well known that acetate esters rapidly decrease with aging, while the ethyl esters of fatty acids stabilize at the chemical equilibrium. On the contrary, branched fatty acid esters increase over time [16].

### 3.2.4. Terpenes and C13-Norisoprenoids

Terpenes are produced in grapes through both the 1-deoxy-d-xylulose-5-phosphate/methylerythritol phosphate (DOXP/MEP) pathway and the mevalonic acid (MVA) pathway [17]. The terpenes produced can then be glycosylated in the berry. During fermentation,

these bound forms are then cleaved by the yeasts releasing the aglycone [18]. Terpenes are an important contributor to floral and citric notes and are found in higher concentrations in aromatic varieties such as Muscat [19,20]. Even if Glera is not a grape variety considered aromatic, a large number of terpenic molecules were identified and quantified in the analyzed wines. Among these terpenes, the one found in higher concentration was linalool, followed by α-terpineol, and linalool oxide. One sample showed a much higher content compared with the others, about 88 μg/L, which was much higher than the odor threshold of 25 μg/L. A wide variability in terpene concentration was observed between the various samples, with differences between minimum and maximum of up to two orders of magnitude. This diversity could be due to various agronomic, viticultural and geographical factors [12,21–23].

Due to their low odor threshold, $C_{13}$-norisoprenoids are very powerful odor compounds. The 3-oxo-α-ionol was quantitatively the most abundant norisoprenoid in the studied wines, followed by TDN, β-damascenone and vitispirane. However, among them, only β-damascenone largely exceeded its odor threshold, potentially contributing to wine aroma as an enhancer of fruity aroma [24]. TDN and vitispirane are formed during wine aging, affecting wine with petrol and camphoraceous aroma notes [25]. Prosecco production regulations state that wines from different vintages can be used for refermentation. However, the low concentrations of both TDN and vitispirane and of bicyclic terpenes such as 1,8-cineole and 1,4-cineole could indicate that the wines were relatively young wines or that they did not undergo thermal stress during storage [26].

### 3.2.5. $C_6$-Alcohols

$C_6$-alcohols are characterized by a herbaceous and leafy aroma [27]. They are formed during berry crushing by the oxidative cleavage of unsaturated fatty acids catalyzed by grape enzymes [28]. The concentration found in the wine samples was lower than the odor thresholds, suggesting a limited contribution of this class of compounds to Prosecco aroma. The variation of $C_6$-alcohols was rather limited in the samples analyzed, reaching a maximum variation of 24% in the case of cis-2-hexen-1-ol. The total $C_6$-alcohols content and the ratio between the cis- and trans-3-hexen-1-ol are strongly influenced by the grape variety [29–31], with the low variability observed in the samples suggesting a homogeneity of the grape variety used. In any case, the 15% of grapes from non-Glera varieties allowed by the production requirements did not have a considerable impact on the $C_6$-alcohols profile.

### 3.2.6. Volatile Sulfur Compounds

Low molecular weight volatile sulfur compounds (VSC) are potent aroma compounds due to their low odor threshold. At high concentration they are responsible for the unpleasant odors of onion, rotten egg and rubber [32], but at lower concentration, some of the VSC can positively contribute to wine aroma [33,34]. The most abundant VSC was ethyl thioacetate, a compound characterized by rotten vegetable aroma [35], which has been suggested to form methanthiol from acid hydrolysis during wine aging [36]. In general, the VSC concentrations observed were quite low in terms of ability to play a major role in the aroma of the sample wines.

### 3.2.7. Benzenoids

Benzenoids are generally characterized by spicy aroma notes. The concentration of benzenoids in Prosecco wines was quite low, suggesting that they did not play a major role in these wines.

### *3.3. Odor Activity Values*

In order to assess the compounds that contribute most to Prosecco aroma, odor activity values (OAV) were calculated for each compound. Compounds with an OAV > 1 or with a concentration greater than their olfactory threshold were considered to have an impact on the aroma of the wine. Table 4 shows the compounds which showed OAV values > 1

in at least one sample. The data showed that the compounds with higher OAV values were mainly the esters. The three highest were, in descending order, ethyl octanoate, ethyl hexanoate and isoamyl acetate. Ethyl butanoate and ethyl 3-methylbutanoate also showed mean values above 1. This suggested that Prosecco wines are mainly characterized by the fruity notes contributed by fermentative esters. Among the other compounds with OAV > 1, fermentative compounds were found such as octanoic acid, hexanoic acid, 3-methylbutanoic acid, isoamyl alcohol, and β-damascenone, which can act as an aroma exhauster of fruity notes. Compounds responsible for aging notes such as TDN and TPB, kerosene, and tobacco, in some samples, could have contributed to the overall odor by affecting the fruity notes characterizing Prosecco. VSC compounds such as dimethyl trisulfide and dimethyl sulfide, despite low concentrations, could also have played a role in some samples, showing mean OAV values of 3.7 and 0.7, respectively.

**Table 4.** Odor activity values of aroma compounds in Prosecco wines with at least one sample showing OAV > 1.

| Volatile Compound | Odor Descriptor | Odor Threshold (μg/L) [1] | Mean | Min | Max |
|---|---|---|---|---|---|
| Ethyl octanoate | Fruity | 5 | 148.2 | 83.3 | 228.6 |
| Ethyl hexanoate | Fruity | 14 | 79.8 | 59.3 | 123.6 |
| Isoamyl acetate | Banana | 30 | 52.5 | 2.0 | 173.3 |
| β-Damascenone | Quince | 0.05 | 30.5 | 1.4 | 104.6 |
| Octanoic acid | Rancid | 500 | 17.6 | 0.3 | 20.4 |
| Hexanoic acid | Rancid | 420 | 14.8 | 10.4 | 18.2 |
| Ethyl butanoate | Fruity | 20 | 13.5 | 3.0 | 21.3 |
| 3-Methylbutanoic acid | Rancid | 33 | 7.9 | 5.2 | 10.7 |
| Ethyl 3-methylbutanoate | Red fruits | 3 | 5.6 | 0.0 | 17.1 |
| Dimethyl trisulfide | Onion | 0.2 | 3.7 | 0.3 | 19.1 |
| Isoamyl alcohol | Vinous | 30,000 | 3.1 | 2.4 | 3.7 |
| TDN | Kerosene | 2 | 1.0 | 0.0 | 4.0 |
| TPB | Tobacco | 0.04 | 0.9 | 0.0 | 3.8 |
| DMS | Truffle, Red fruits | 10 | 0.7 | 0.3 | 1.8 |
| Phenylethyl acetate | Rose | 250 | 0.6 | 0.1 | 2.3 |
| Ethyl-2-methylbutanoate | Red fruits | 18 | 0.5 | 0.0 | 1.5 |
| Linalool | Floral | 25 | 0.5 | 0.0 | 3.5 |

[1] Data from: [19,20,37–41].

### 3.4. Effect of the PDO on Volatile Profile

In order to evaluate the influence of the area of origin on the aromatic composition of the wines, the data were subjected to non-parametric analysis by Kruskal–Wallis test ($\alpha$ = 0.1). Even if the dataset was not balanced among the production areas, some observations could be obtained. Samples from the three areas were quite similar for most compounds, with only 24 compounds (11 terpenes, 6 VSC, 2 norisoprenoids, 2 benzenoids and 3 fermentative compounds) showing significant differences (Table 3). The 11 terpenes were nerol, geraniol, α-phellandrene, limonene, γ-terpinene, p-cymene, α-terpinene, β-myrcene, 3-carene, β-pinene, and rose oxide. These terpenes were generally found in higher concentration in Asolo samples, except for nerol, which was higher in Valdobbiadene samples. In fact, it has been observed that terpenes are influenced by the origin of the grapes as well as by viticultural practices [12,42,43]. Even if the two areas are very close, the historical series of weather data indicates differences, for example, in terms of annual temperatures, with Asolo tending to be slightly colder "https://www.arpa.veneto.it/dati-ambientali/open-data (accessed on accessed on 3 March 2023)". However, it should be remembered that the hilly conformations can lead to considerable microclimatic differences even within the PDO itself. In this way, they could actively contribute to the geographical typicality of the wines. Among these terpenes, only nerol

and geraniol were linear monoterpene alcohols characterized by floral notes, the rest were cyclic monoterpenoids that could contribute to citric, turpentine and pine aromas.

Asolo wines also showed the highest content of the two isoprenoids TPB and TDN. TDN concentration in wine has been reported to be influenced not only by wine age, but also by grape sun exposure, pH, and storage temperature [25,44]. TPB is characterized at low concentration by tobacco aroma notes, while at high concentration it has been described as a geranium-like aroma. The concentrations of TPB were quite low, however, thanks to the very low olfactory threshold (0.04 ug/L) [39], TPB could play a sensorial role and characterize the aroma of the wines of the Asolo area. TPB has been shown to increase during aging [26], moreover, it can react with tannins; for this reason it is more present in white wines or wines with a low concentration of polyphenols [45].

Six VSC were significantly different, showing, in general, slightly higher but significantly different concentrations in Asolo wines (Figure 1). The presence of these compounds can negatively contribute to wine aroma with the smell of onions, rotten egg, and truffle. Many factors promote the formation of VSCs, such as residual sulfur-containing pesticides, yeast strain used for fermentation, development of microorganisms, and temperature of fermentation [35,46,47].

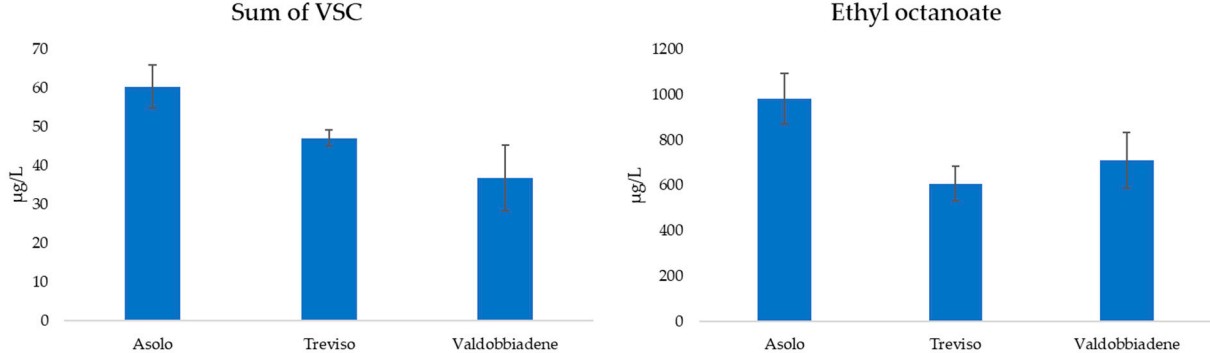

**Figure 1.** Concentration of the sum of VSC and ethyl octanoate in wines from Asolo, Treviso and Valdobbiadene.

With regard to the fermentative compounds, ethyl octanoate, hexanoic acid and phenylethyl alcohol showed statistically significant differences. Wine samples from Treviso DOC showed a lower amount of hexanoic acid and phenylethyl alcohol. Ethyl octanoate was found to be more concentrated in Asolo wines. It should be noted that, although not significant, the content of ethyl hexanoate also tended to be higher in Asolo samples, while no significant differences were observed for acetic esters and branched ethyl esters.

The benzenoids 4-vinylguaiacol and benzaldehyde were found at higher levels in Valdobbiadene wines. However, the concentrations detected did not appear to be such as to involve a real olfactory contribution.

### 3.5. Sorting Task Analysis

Wines were submitted to sorting task analysis in order to assess the existence of sensory groups based on their odor similarities. This approach has already been used to establish the existence of odor profiles associated with different variables, including grape variety, grape origin and yeast strain [10,13,43,48]. In this study, we were interested in verifying the existence of sensory groups attributable to the three different PDO appellations. The data obtained from the sorting task were entered into individual matrices for each judge, these matrices were aggregated in a single matrix and submitted to hierarchical cluster analysis (HCA). The results, shown in Figure 2, showed that the replicate wines were projected in the same cluster (sample no. 11), very close to each other, meaning that the panel was reproducible. The results showed that the Prosecco wines were divided into two groups of sensory similarity. The first group, (A), consisted of 11 wines, the second group, (B),

of all other wines. The sensory groups were formed so that only part of the clustering could be attributed to the different PDOs. In fact, group A was almost entirely formed by Valdobbiane PDO and group B was formed by a mix of the three denominations with the exception of sample no. 1 which was derived, instead, from the DOC Treviso. These observations indicated that despite a strong segmentation according to different PDOs, a specific olfactory space did not exist for each of the two clusters. The volatile compounds characterizing the sensory clusters were then identified by means of Mann–Whitney test. Twenty-two volatile compounds showed significant differences ($\alpha$ = 0.1) across the two clusters (Table 5). Significant compounds were mostly terpenoids (8), VSC (4), esters acetate and ethyl esters (4), $C_6$-alcohols (2), and one each of norisoprenoid, sesquiterpenoid, fatty acid and benzenoid.

Cluster A was richer in esters such as isoamyl acetate, phenyl ethyl acetate and ethyl butanoate, that contributed to the fruity parameter, with acetates in cluster A showing about twice the content of cluster B. Cluster A was also richer in benzaldehyde and octanoic acid, with only octanoic acid largely exceeding the olfactory threshold and able to actively contribute to the aroma of the wines. Cluster B was richer in VSC, dimethyl sulfide, diethyl sulfide, dimethyl disulfide, and diethyl disulfide, all of which are characterized by the unpleasant aroma notes of onion, garlic, and sulfur. Moreover, cluster B showed a higher level of cyclic terpenoids such as cis-linalool oxide, terpinene-1-ol, 1,4-cineole, p-cymene, and p-menthane-1,8-diols, and in the norisoprenoids, vitispirane; these compounds are characterized by odors of turpentine, resin, and camphor, which could have a negative impact—even if present at concentrations below the olfactory threshold—on the olfactory quality of cluster B wines. An average higher concentration of TDN was observed in cluster B, contributing to an aging aroma. A pleasant aroma that characterized cluster B was geraniol with its floral notes.

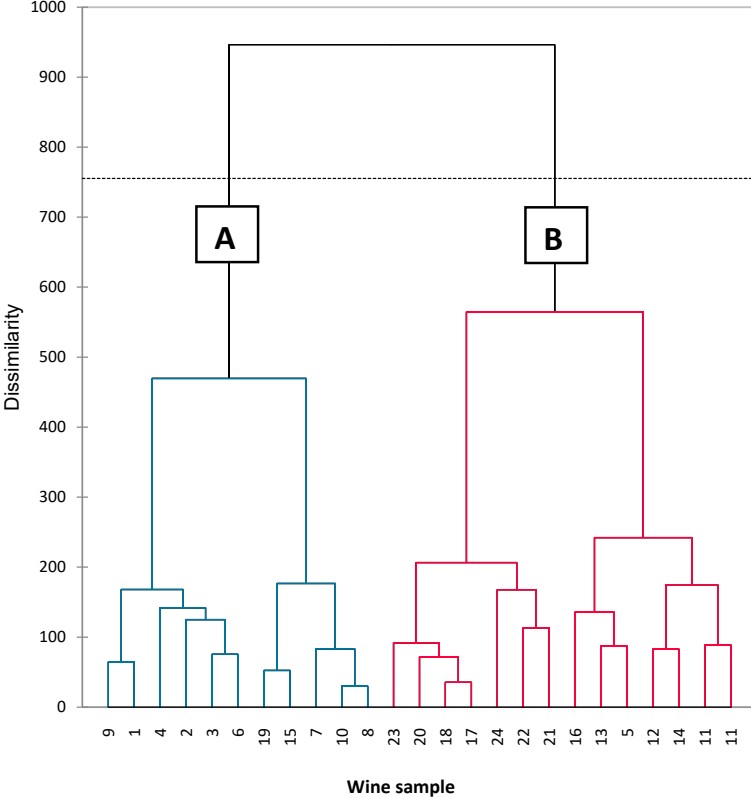

**Figure 2.** Agglomerative hierarchical clustering of the sorting task scores; the numbers indicate Prosecco wine samples, the letters A and B indicate clusters. The dashed line indicates the significance threshold.

**Table 5.** Concentration (µg/L) of free volatile compounds in cluster 1 and cluster 2 wine samples. Odor threshold, mean, standard deviation (SD), statistical significance (*p*-value), and level of significance ($p < 0.1$ *; $p < 0.05$ **; $p < 0.01$ ***) are shown.

| Compound | Odor Threshold (µg/L) [1] | Cluster A | | Cluster B | | | |
|---|---|---|---|---|---|---|---|
| | | Mean (µg/L) | SD | Mean (µg/L) | SD | *p*-Value | Level |
| Carbon disulfide | 38 | 14.2 | ±3.38 | 15.83 | ±4.04 | 0.277 | |
| Dimethyl sulfide | 10 | 4.91 | ±1.58 | 8.89 | ±3.81 | 0.002 | *** |
| Diethyl sulfide | 18 | 0.64 | ±0.38 | 1.7 | ±0.6 | <0.0001 | *** |
| Dimethyl disulfide | 30 | 0.71 | ±0.86 | 2.42 | ±1.18 | 0.005 | *** |
| Ethyl thioacetate | 40 | 13.89 | ±8.39 | 18.27 | ±9.39 | 0.207 | |
| Diethyl disulfide | 4.3 | 0.09 | ±0.1 | 0.41 | ±0.3 | 0.000 | *** |
| Dimethyl trisulfide | 0.2 | 0.34 | ±0.34 | 1.1 | ±1.4 | 0.744 | |
| Sum of VSC | | 34.77 | ±7.99 | 48.61 | ±10.65 | 0.006 | *** |
| 1-Butanol | 150,000 | 89.22 | ±22.36 | 84.44 | ±22.61 | 0.531 | |
| Isoamyl alcohol | 30,000 | 93,446.18 | ±8421.84 | 92,598 | ±11,058.2 | 0.955 | |
| Phenylethyl alcohol | 14,000 | 5701.79 | ±1110.78 | 5148.49 | ±1176.82 | 0.277 | |
| Benzyl alcohol | | 88.04 | ±80.2 | 63.96 | ±19.37 | 0.776 | |
| Methionol | 1000 | 176.98 | ±52.42 | 203.58 | ±66.71 | 0.331 | |
| Sum of higher alcohol | | 99,412.99 | ±9275.54 | 98,014 | ±12,077.15 | 0.955 | |
| 1-Hexanol | 8000 | 844.62 | ±143.82 | 1002.5 | ±166.54 | 0.022 | ** |
| cis-3-Hexen-1-ol | 1000 | 14.07 | ±2.64 | 16.66 | ±3.42 | 0.040 | ** |
| trans-3-Hexen-1-ol | 400 | 138.91 | ±21.85 | 143.11 | ±27.59 | 0.569 | |
| cis-2-Hexen-1-ol | | 13.37 | ±3.84 | 12.93 | ±2.51 | 0.776 | |
| Sum of C6 alcohols | | 1010.96 | ±147.9 | 1175.2 | ±173.47 | 0.018 | ** |
| Isoamyl acetate | 30 | 2164.76 | ±1270.04 | 1073.78 | ±617.39 | 0.041 | ** |
| n-Hexyl acetate | 1500 | 92.57 | ±48.72 | 52.88 | ±31.71 | 0.106 | |
| Phenethyl acetate | 250 | 196.24 | ±135.08 | 115.82 | ±55.45 | 0.063 | * |
| Sum of acetates | | 2453.57 | ±1442.73 | 1242.49 | ±696.56 | 0.041 | ** |
| Ethyl acetate | 12,264 | 59.72 | ±25.55 | 45.66 | ±19.15 | 0.228 | |
| Ethyl butanoate | 20 | 290.62 | ±48.25 | 254.11 | ±75.9 | 0.082 | * |
| Ethyl hexanoate | 14 | 1185.59 | ±272.96 | 1059.89 | ±184.97 | 0.277 | |
| Ethyl octanoate | 5 | 743.37 | ±102.26 | 738.53 | ±203.34 | 0.820 | |
| Ethyl decanoate | 200 | 95.33 | ±35.99 | 85.8 | ±32.16 | 0.733 | |
| Sum of ethyl esters of straight-chain fatty acids | | 2374.63 | ±388.97 | 2184 | ±433.82 | 0.277 | |
| Ethyl 2-methylbutyrate | 18 | 5.8 | ±2.66 | 11.06 | ±8.67 | 0.072 | * |
| Ethyl 3-methylbutanoate | 3 | 12.42 | ±5.65 | 20.55 | ±15.02 | 0.167 | |
| Ethyl 3-hydroxybutyrate | | 71.84 | ±11.55 | 60.12 | ±23.21 | 0.277 | |
| Ethyl 2-hydroxyhexanoate | | 0.89 | ±0.27 | 1.06 | ±0.44 | 0.258 | |
| Sum of ethyl esters of branched acid | | 90.96 | ±11.61 | 92.79 | ±33.04 | 0.776 | |
| Ethyl cinnamate | | 5.11 | ±2.29 | 6.05 | ±3.05 | 0.109 | |
| Ethyl lactate | 154,000 | 4386.48 | ±2800.75 | 4538.15 | ±1626.96 | 0.303 | |
| Sum of other esters | | 4391.59 | ±2802.26 | 4544.2 | ±1627.98 | 0.303 | |
| 3-Methylbutanoic acid | 33 | 256.6 | ±44.42 | 261.7 | ±40.81 | 0.865 | |
| Hexanoic acid | 420 | 6435.45 | ±579.8 | 6051.99 | ±861.38 | 0.361 | |
| Octanoic acid | 500 | 9578.26 | ±554.32 | 8178.39 | ±2520.8 | 0.009 | *** |
| Sum of fatty acids | | 16,270.3 | ±1090.48 | 14,492.1 | ±2660.71 | 0.041 | ** |
| cis-Linalool oxide | 3000 | 3.93 | ±1.87 | 8.6 | ±6.76 | 0.018 | ** |
| trans-Linalool oxide | 6000 | 2.49 | ±1.15 | 4.04 | ±3.22 | 0.186 | |
| Linalool | 25 | 11.13 | ±6.03 | 12.52 | ±23.11 | 0.106 | |
| Terpinen-1-ol | | 0.04 | ±0.03 | 0.19 | ±0.33 | 0.010 | ** |
| Terpinen-4-ol | | 1.43 | ±0.82 | 1.04 | ±0.43 | 0.353 | |
| Ho-trienol | 110 | 0.06 | ±0.07 | 0.07 | ±0.17 | 0.264 | |
| α-Terpineol | | 6.8 | ±2.66 | 5.68 | ±3.04 | 0.339 | |

**Table 5.** *Cont.*

| Compound | Odor Threshold (µg/L) [1] | Cluster A | | Cluster B | | *p*-Value | Level |
|---|---|---|---|---|---|---|---|
| | | Mean (µg/L) | SD | Mean (µg/L) | SD | | |
| Nerol | 250 | 3.93 | ±8.97 | 0.99 | ±1.44 | 0.701 | |
| Geraniol | 400 | 1.74 | ±1.45 | 3.25 | ±1.91 | 0.055 | * |
| β-Citronellol | 30 | 3.11 | ±1.84 | 2.25 | ±1.91 | 0.270 | |
| α-Phellandrene | 100 | 0.12 | ±0.06 | 0.12 | ±0.1 | 0.502 | |
| 1,4-Cineole | 0.54 | 0.1 | ±0.04 | 0.22 | ±0.11 | 0.002 | *** |
| 1,8-Cineole | 1.1 | 0.12 | ±0.08 | 0.19 | ±0.1 | 0.129 | |
| Limonene | | 0.17 | ±0.08 | 1.39 | ±4.03 | 0.327 | |
| γ-Terpinene | | 2.19 | ±1.12 | 2.06 | ±1.73 | 0.691 | |
| p-Cymene | | 0.11 | ±0.02 | 0.19 | ±0.1 | 0.005 | *** |
| Terpinolene | | 0.18 | ±0.08 | 0.15 | ±0.09 | 0.484 | |
| p-Menthane-1,8-diol | | 1.92 | ±1.5 | 4.92 | ±4.82 | 0.041 | ** |
| α-Terpinene | | 0.05 | ±0.02 | 0.06 | ±0.04 | 0.853 | |
| β-Myrcene | | 1.09 | ±0.62 | 1.38 | ±1.34 | 0.659 | |
| 3-Carene | | 0.08 | ±0.02 | 0.11 | ±0.07 | 0.209 | |
| β-Pinene | | 0.11 | ±0.06 | 0.31 | ±0.68 | 0.987 | |
| Rose oxide | | 0.02 | ±0.01 | 0.02 | ±0.01 | 0.318 | |
| Sum of monoterpenoids | | 40.92 | ±15.95 | 49.76 | ±28.25 | 0.277 | |
| β-Damascenone | 0.05 | 1.96 | ±1.39 | 1.15 | ±0.78 | 0.172 | |
| Vitispirane 1 | | 0.6 | ±0.26 | 1.47 | ±1.17 | 0.015 | *** |
| Vitispirane 2 | | 0.56 | ±0.22 | 0.89 | ±0.64 | 0.130 | |
| 1-(2,3,6-Trimethylphenyl)-buta-1,3-diene (TPB) | 0.04 | 0.03 | ±0.02 | 0.05 | ±0.05 | 0.626 | |
| 1,1,6-Trimethyl-1,2-dihydronapthalene (TDN) | 2 | 0.92 | ±0.34 | 2.93 | ±2.1 | 0.002 | *** |
| 3-Oxo-α-ionol | | 2.48 | ±0.69 | 2.67 | ±1.02 | 0.608 | |
| Sum of norisoprenoids | | 4.58 | ±1.14 | 8 | ±3.63 | 0.005 | *** |
| Nerolidol | | 2.36 | ±1.22 | 1.16 | ±0.56 | 0.007 | *** |
| Bisabolol | | 1.26 | ±0.18 | 1.59 | ±1.11 | 0.721 | |
| Sum of sesquiterpenoids | | 3.61 | ±1.25 | 2.75 | ±1.17 | 0.055 | * |
| 4-Vinylguaiacol | | 8.02 | ±2.37 | 6.52 | ±1.66 | 0.207 | |
| Vanillin | 200 | 2.56 | ±0.35 | 4.59 | ±6.35 | 0.579 | |
| Methyl vanillate | 3000 | 4.45 | ±0.82 | 3.94 | ±0.38 | 0.172 | |
| Ethyl vanillate | 990 | 1.92 | ±0.63 | 1.65 | ±0.54 | 0.353 | |
| Methyl salycilate | 38 | 3.73 | ±1.37 | 3.1 | ±1.39 | 0.450 | |
| Benzaldehyde | | 8.11 | ±3.49 | 2.43 | ±2.32 | 0.001 | *** |
| Sum of benzenoids | | 28.79 | ±6.16 | 22.23 | ±8 | 0.015 | ** |

[1] Data from: [19,20,37–41].

## 4. Conclusions

This research provides an extensive chemical characterization of the volatile compound profile of Prosecco wine. The aroma of Prosecco wines was characterized mainly by ethyl esters of straight-chain fatty acids and acetates. Other compounds that characterized the volatile composition of Prosecco wine were the fermentative compounds, such as fatty acids, phenylethyl alcohol, and the sulfur-containing compounds dimethyl trisulfide and dimethyl sulfide. Monoterpenes could have, in some samples, contributed to the floral notes, however, in general, their concentration remained lower than their odor threshold. Compositional differences were observed between the three PDOs analyzed, with the Asolo wines generally richer in terpenes, norisoprenoids and sulfur compounds. Valdobbiadene was mainly characterized by benzenoids, while Treviso was mainly characterized by hexanoic acid and phenylethyl alcohol. Sensory analysis performed by mean of sorting task methodology which indicated that the wines were grouped to form two clusters not perfectly matching with the three PDOs considered. The two clusters were characterized by

22 compounds; in particular, cluster A was characterized by a higher esters content while cluster B was most distinguished by higher levels of VSC, cyclic terpenes and TDN. This work helps to characterize the aromatic profile of Prosecco wine. The results may be of help to the winemaker in promoting the aromatic characteristics of Prosecco, in particular by paying attention to the production of fermentative compounds with fruity notes such as esters, and to the enhancement of fruity floral notes linked to terpenes and norisoprenoids; for example, in the cellar, by the choice of yeast strains, nitrogen nutrition, fermentation temperature; or in the vineyard, by decreasing the formation of precursors of TDN such as limiting grapes' sun exposure, or through foliar nitrogen nutrition.

**Author Contributions:** Conceptualization, D.S. and G.L.; methodology, M.U., D.S. and G.L.; validation, D.S.; formal analysis, G.L. and M.B.; data curation, G.L., D.S. and M.U.; writing—original draft preparation, D.S.; writing—review and editing, M.U. and G.L.; visualization, D.S.; project administration, D.S.; funding acquisition, D.S., A.B., A.D.C., V.Z. All authors have read and agreed to the published version of the manuscript.

**Funding:** The work was funded by Joint Project 2019 developed by the University of Verona and Masi Agricola S.p.a., JPVR19WJAC titled "Influenza di pratiche enologiche sull'aroma di vini Prosecco, studio dell'utilizzo di enzimi durante la rifermentazione in autoclave".

**Institutional Review Board Statement:** Not applicable.

**Informed Consent Statement:** Not applicable.

**Data Availability Statement:** The data presented in this study are available on request from the corresponding author.

**Conflicts of Interest:** Davide Slaghenaufi, Matteo Borgato, Giovanni Luzzini and Maurizio Ugliano declare no conflict of interest. Anita Boscaini, Vittorio Zandonà and Andrea Dal Cin work at Masi Agricola, funder of the project. They had no role in the design of the study; in the collection, analyses, or interpretation of data and in the writing.

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
