# Peer review of "Characterization of the Aroma Profile of Commercial Prosecco Sparkling Wines"

_applsci, doi:10.3390/app13063609_

Round 1

Reviewer 1 Report

The authors did an aroma and chemical descriptive study on what a specific group of Prosecco wines from three regions are driven by. The importance of the information for the industry and winemaking process is critical to further improve the quality of consumer preference of style as this style of sparkling wine is growing world-wide from a consumption point.

The Introduction section could have given some background on typical viticultural practices and winemaking protocols to give context and also relating to the origin or differences of compounds observed. also context to why the price bracket was chosen?

The discussion of the results is limited and can be improved by giving some explanation to possible results obtained related to compounds or correlate with winemaking practice in the cellars in specific regions.

General comments:

52 ... the volatile aroma profile of sparkling wines...

98 Table with a capital letter

Table 1 is there no Brut in Asolo in the price category or do they not produce Brut in region?

2.6 section relates mostly to Table 2, many of the compounds are not listed in this section and how they were measured ? in table PAN, where is YAN that is mentioned in 2.6. was ethanol not measured or do all wines have a strict alcohol level for Prosecco, then please mention range?

Table 2 for the reader maybe give the sugar level that classifies a Brut and extra dry to give perspective to the sugar levels

226: ... higher alcohols are generally characterized by solvent odors? hey does not make sense

239: if ethyl lactate is one of the major esters does this mean that all wines have completed MLF or is it inoculated in the basewine?

246 acetic acid esters or acetate esters?

340-345 to discuss this results is there any data from the regions that would confirm hypothesis such as climate or viticultural practices etc.

358-360 Any data from region to correlate with this statement?

With only two groups being formed with sorting on aroma, what does mean for the PDO system or to relate it back to your aim the influence of geographic origin?

Any conclusion on viticulture or winemaking practices that would aid the wine style in future to link it better to what the consumer wants?

Author Response

1#

The authors did an aroma and chemical descriptive study on what a specific group of Prosecco wines from three regions are driven by. The importance of the information for the industry and winemaking process is critical to further improve the quality of consumer preference of style as this style of sparkling wine is growing world-wide from a consumption point.

The Introduction section could have given some background on typical viticultural practices and winemaking protocols to give context and also relating to the origin or differences of compounds observed. also context to why the price bracket was chosen?

  • Additional information regarding Prosecco viticulture and enology has been added in the introduction. Lines 40-48
  • Price explanation was added in section 2.2. to the line 111-112

The discussion of the results is limited and can be improved by giving some explanation to possible results obtained related to compounds or correlate with winemaking practice in the cellars in specific regions.

The PDOs analyzed are very close in terms of distance, furthermore to regulate the oenological practices do not vary between the areas analysed. Basically there is no diversity of winemaking practices between the areas due to local tradition or disciplinary. The diversity in oenological techniques can be traced back to company choices. Furthermore, it is not possible to specifically obtain the winemaking protocols of the companies used in the study, for these reasons no further considerations have been made from an oenological point of view.

General comments:

52 ... the volatile aroma profile of sparkling wines...

  • correction done. Line 63

98 Table with a capital letter

  • correction done. Line 115

Table 1 is there no Brut in Asolo in the price category or do they not produce Brut in region?

  • Among the wine samples available in the local large-scale distribution at the time of the work, Asolo in the Brut type was not found.

2.6 section relates mostly to Table 2, many of the compounds are not listed in this section and how they were measured ? in table PAN, where is YAN that is mentioned in 2.6. was ethanol not measured or do all wines have a strict alcohol level for Prosecco, then please mention range?

YAN has been added to table 2. The compounds reported in table 2 which had not been mentioned in materials and methods were measured by means of special enzymatic and colorimetric kits with the Biosystem automatic analyzer. The missing compounds have been added in section 2.6. Line 204, 207

Data concerning alcool has been added in table 2 and the method has been reported in section 2.6 of material and methods

Table 2 for the reader maybe give the sugar level that classifies a Brut and extra dry to give perspective to the sugar levels

The sugar level of brut and extra dry has been added to table 2

226: ... higher alcohols are generally characterized by solvent odors? hey does not make sense

The sentence has been rewritten. Lines 265-267

239: if ethyl lactate is one of the major esters does this mean that all wines have completed MLF or is it inoculated in the basewine?

as the wines all still show at least 1.3 g/L of malic acid and only up to 0.6 g/L of lactic acid it does not appear that the wines have undergone malolactic fermentation. However, it has been observed that ethyl lactate increases in wines subjected to second fermentation and further increases with sur lie (Torrens et al., J. Agric. Food Chem. 2010, 58, 4, 2455–2461)( Bernardo, et al.. Foods 2022, 11, 1529.https://doi.org/10.3390/foods11111529)

246 acetic acid esters or acetate esters?

Correction done. Line 298

340-345 to discuss this results is there any data from the regions that would confirm hypothesis such as climate or viticultural practices etc.

Regional historical series indicate that temperatures in the PDO of Asolo tend to be cooler than in Valdobbiadene even if the latter is located at a higher altitude. These data should be considered with caution as there are large microclimatic differences due to the pre-mountain hilly environment of the two PDOs. A sentence has been added reporting these considerations. Lines 394-398

358-360 Any data from region to correlate with this statement?

We have no data to support a particular thesis for this reason all the factors that we know can influence the level of VSC have been reported as possible.

With only two groups being formed with sorting on aroma, what does mean for the PDO system or to relate it back to your aim the influence of geographic origin?

The two clusters were formed by entropy truncation at this significance level. However, if we look inside clusters, we can see further subclusterizations. In particular within cluster B, we have observed that according to the confidence level we choose, almost all Asolo samples are grouped together indicating an olfactory similarity. The Treviso PDO, on the other hand, is very broad and includes the other two PDOs, for this reason it is difficult to distinguish in a separate cluster. Since the focus of the project was not to identify the differences between geographical areas we decided to be more cautious with the conclusions.

Any conclusion on viticulture or winemaking practices that would aid the wine style in future to link it better to what the consumer wants?

A sentence has been added in the conclusion section. Lines 497-504

Reviewer 2 Report

The authors in their present study have analysed a great quantity of volatile compounds from Prosecco wines by GC-MS analysis in order to provide an extensive chemical and sensory characterization as well as determine if the geographic origin has an influence on the analysed profiles. More precisely 5 wines were from the Treviso region, 4 from the Asolo and 16 from Valdobbiadene. According to their results compositional differences were observed between the three PDOs analysed while the sensory analysis by sorting task tested revealed the existence of two main clusters. The amount of chemical analysed is great giving a complete aromatic profile, whatsoever the wine samples of each region (especially Asolo and Treviso) are not adequate in order to make any conclusion on the geographic impact. So it should be paid more attention to the the way the results are presented and how the conclusion is made. From a general point of view the work is interesting.

More detailed comments:

Line 15

‘’in quantitative terms ‘’ is not the correct expression

Line 16

What is VSC?

Line 17

Prosecco or prosecco?

Line 20

Add the word wine after Prosecco

Line 23

Correct your expression

Line 35

Denominazione di Origine Controllata…write it pls

Line 35-40

I think you should explain more the winemaking method of Prosecco wine especially the Charmant method

Line 44-45

Attention to your expression

Line 48

DOCG?

Line 49-54

Attention to your expression

Line 55-57

Attention to your expression

Line 65

‘’Some articles investigate the mineral composition of wine for authentication  purposes’’ reference?

Material and methods

Is there any technical repetition?

Line 203

The same market? So the same conservation conditions??

Line 208

Any comment for lactic acid?

For max free SO2?

Is normal the quantity of tartaric acid?

Table 2

What do you mean by ‘’primary’’ amino nitrogen?

Line 331

It would be nice to add a figure for this section

Line 214

No comments for 3.2.1 and 3.2.2?

Line 346

Why due to the base wine?

Author Response

The authors in their present study have analysed a great quantity of volatile compounds from Prosecco wines by GC-MS analysis in order to provide an extensive chemical and sensory characterization as well as determine if the geographic origin has an influence on the analysed profiles. More precisely 5 wines were from the Treviso region, 4 from the Asolo and 16 from Valdobbiadene. According to their results compositional differences were observed between the three PDOs analysed while the sensory analysis by sorting task tested revealed the existence of two main clusters. The amount of chemical analysed is great giving a complete aromatic profile, whatsoever the wine samples of each region (especially Asolo and Treviso) are not adequate in order to make any conclusion on the geographic impact. So it should be paid more attention to the the way the results are presented and how the conclusion is made.

We agree with the reviewer for this reason no further considerations have been made on the areas of origin. A consideration on the climatic differences of Asolo and Valdobbiadene has been inserted as a request of reviewer 1, at line 394-398

From a general point of view the work is interesting.

More detailed comments:

Line 15

‘’in quantitative terms ‘’ is not the correct expression

The sentence has been rewritten. Line 15

Line 16

What is VSC?

VSC definition has been added. Line 17

Line 17

Prosecco or prosecco?

Because Prosecco is the name of the wine, it has been corrected throughout the text by capitalizing it

Line 20

Add the word wine after Prosecco

The word “wine” has been added after Prosecco. Line 21

Line 23

Correct your expression

The sentence has been rewritten. Now line 23

Line 35

Denominazione di Origine Controllata…write it pls

Correction done line 38

Line 35-40

I think you should explain more the winemaking method of Prosecco wine especially the Charmant method

Additional information regarding Prosecco viticulture and enology has been added in the introduction. Lines 46-49

Line 44-45

Attention to your expression

The sentence has been changed. Line 53

Line 48

DOCG?

The acronym DOCG has been detailed to the line 43

Line 49-54

Attention to your expression

The sentence has been changed. Line 60-63

Line 55-57

Attention to your expression

The sentence has been changed. Line 66-68.

Line 65

‘’Some articles investigate the mineral composition of wine for authentication  purposes’’ reference?

Reference has been added at line 77

Material and methods

Is there any technical repetition?

There was no analytical repetition as the methods have already been validated.

No technical replicas were made, on several bottles of the same wine, because it would not have corresponded to what the jury had tasted.

Line 203

The same market? So the same conservation conditions??

That's right the wines came from the same supermarket, the storage conditions were the same what could vary was the time of arrival at the store and the transport from the cellar to the store

Line 208

Any comment for lactic acid?

For max free SO2?

Is normal the quantity of tartaric acid?

 Comment on lactic acid has been added. Line 247-248

Comment on free SO2 has been added. Line 248-251

Regarding tartaric acid we do not have other data relating to prosecco to evaluate whether they are normal or not

Table 2

What do you mean by ‘’primary’’ amino nitrogen?

As reported by Waterhouse in Understanding wine chemistry: “… primary α‐amino acids the amine group is bonded to only one carbon (R–NH2), and the acid and amine groups form bonds to the same carbon…”

Line 331

It would be nice to add a figure for this section

A figure has been added as fig.1.

Line 214

No comments for 3.2.1 and 3.2.2?

Additional comments have been added to the sections 3.2.1 and 3.2.2.

Line 346

Why due to the base wine?

 On line 346 I can't find the reference to the base wine. If instead reviewer refers to line 246, the hypotheses are that either it is the commercial Prosecco wine (refermented and bottled) that is old and therefore loses the acetic esters, or it is already the base wine that is a bit old and has already lost the acetic esters before refermentation.

Reviewer 3 Report

The work described in the manuscript fills a gap in the knowledge of constituents of prosecco wine, especially regarding sensory constituents and their variation according to geographical location of vineyards that produce the wine. I don't know of any previous work that gives detailed coverage of the latter point. The methodology (SPME, SPE and GC) used is up to date, and while it is not innovative, I cannot see any way in which it could be improved. The conclusions drawn from the results are very reasonable and I do not feel that the work is "over-referenced".

I recommend acceptance of this manuscript, after considetation of my comments and suggestions.

Line 224. Delete "the" before the names of these constituents
Line 225. Insert "cooked" before "potatoes"
Line 226. Delete "hey"
Line 292. "Rubber" is a common sensory descriptor, rather than "tire"
Line 333. Replace "by" with "of". Also, it should be "Kruskal-Wallis" (with hyphen)
Line 338. "Terpinen" should be "Terpinene"
Line 338, 9. Begin this sentence with "Many factors promote the formation of VSCs, like...." (then continue as already written)
Table 1. Details of "brut" and "extra dry" could be given in a footnote (e.g., "brut and extra dry refer to residual sugar concentrations of 0-12 g/L and 12-17 g/L, respectively")
Table 3. Check consistency of lower and upper case (e.g., cis-3-Hexen-1-ol and cis-2-hexen-1-ol)
Tables 3, 4, and 5. Check consistency of use of comma or full stop to indicate decimal point (comma is used in the tables, but full stop elsewhere)
Table 5. "Terpinen" should be "Terpinene"

Author Response

#3

The work described in the manuscript fills a gap in the knowledge of constituents of prosecco wine, especially regarding sensory constituents and their variation according to geographical location of vineyards that produce the wine. I don't know of any previous work that gives detailed coverage of the latter point. The methodology (SPME, SPE and GC) used is up to date, and while it is not innovative, I cannot see any way in which it could be improved. The conclusions drawn from the results are very reasonable and I do not feel that the work is "over-referenced".

I recommend acceptance of this manuscript, after considetation of my comments and suggestions.

Line 224. Delete "the" before the names of these constituents

Correction done. “the” before constituents name has been delated. Line 265

Line 225. Insert "cooked" before "potatoes"

Cooked has been added. Line 266

Line 226. Delete "hey"

“hey” has been delated. Line 267

Line 292. "Rubber" is a common sensory descriptor, rather than "tire"

“tire” has been changed with “rubber”. Line 344

Line 333. Replace "by" with "of". Also, it should be "Kruskal-Wallis" (with hyphen)

Corrections done. Now line 385

Line 338. "Terpinen" should be "Terpinene"

Corrections done at line 390

Line 338, 9. Begin this sentence with "Many factors promote the formation of VSCs, like...." (then continue as already written)

Correction done now at lines 414-415

Table 1. Details of "brut" and "extra dry" could be given in a footnote (e.g., "brut and extra dry refer to residual sugar concentrations of 0-12 g/L and 12-17 g/L, respectively")

Details have been added to table 1

Table 3. Check consistency of lower and upper case (e.g., cis-3-Hexen-1-ol and cis-2-hexen-1-ol)

Correction done in table 3

Tables 3, 4, and 5. Check consistency of use of comma or full stop to indicate decimal point (comma is used in the tables, but full stop elsewhere)

the correction has been made, the point has been placed in the tables to indicate the decimal

Table 5. "Terpinen" should be "Terpinene"

Corrections have been made in table 5

Reviewer 4 Report

It would be good to list the sources of literature in certain sub-chapters of the chapter Materials and research methods, as shown in the example in the paper.

Author Response

Apologies to the reviewer but I don't understand the suggestion. What is meant?

Round 2

Reviewer 1 Report

The authors are thanked for considering inputs from reviewers to improve the paper and content.

Reviewer 2 Report

No comments. Thank you for your answers